# “Dividing and Conquering” and “Caching” in Molecular Modeling

**DOI:** 10.3390/ijms22095053

**Published:** 2021-05-10

**Authors:** Xiaoyong Cao, Pu Tian

**Affiliations:** 1School of Life Sciences, Jilin University, Changchun 130012, China; caoxy17@mails.jlu.edu.cn; 2School of Artificial Intelligence, Jilin University, Changchun 130012, China

**Keywords:** molecular modeling, multiscale, coarse graining, molecular dynamics simulation, Monte Carlo simulation, force fields, neural network, many body interactions, sampling, local sampling, local free energy landscape, generalized solvation free energy

## Abstract

Molecular modeling is widely utilized in subjects including but not limited to physics, chemistry, biology, materials science and engineering. Impressive progress has been made in development of theories, algorithms and software packages. To divide and conquer, and to cache intermediate results have been long standing principles in development of algorithms. Not surprisingly, most important methodological advancements in more than half century of molecular modeling are various implementations of these two fundamental principles. In the mainstream classical computational molecular science, tremendous efforts have been invested on two lines of algorithm development. The first is coarse graining, which is to represent multiple basic particles in higher resolution modeling as a single larger and softer particle in lower resolution counterpart, with resulting force fields of partial transferability at the expense of some information loss. The second is enhanced sampling, which realizes “dividing and conquering” and/or “caching” in configurational space with focus either on reaction coordinates and collective variables as in metadynamics and related algorithms, or on the transition matrix and state discretization as in Markov state models. For this line of algorithms, spatial resolution is maintained but results are not transferable. Deep learning has been utilized to realize more efficient and accurate ways of “dividing and conquering” and “caching” along these two lines of algorithmic research. We proposed and demonstrated the local free energy landscape approach, a new framework for classical computational molecular science. This framework is based on a third class of algorithm that facilitates molecular modeling through partially transferable in resolution “caching” of distributions for local clusters of molecular degrees of freedom. Differences, connections and potential interactions among these three algorithmic directions are discussed, with the hope to stimulate development of more elegant, efficient and reliable formulations and algorithms for “dividing and conquering” and “caching” in complex molecular systems.

## 1. Introduction

The impact of molecular modeling in scientific research is clearly embodied by the number of publications. Statistics from a Web of Science (www.webofknowledge.com (accessed on 15 February 2021)) search with various relevant key words is listed in Table 1. However, despite widespread applications of molecular modeling, we remain far from accurately predicting and designing molecular systems in general. Further methodological development is highly desired to tap its full potential. Historically, molecular modeling has been approached from a physical or application point of view, and numerous excellent reviews are available in this regard [1,2,3,4,5,6,7,8,9,10,11,12,13,14,15,16]. From an algorithmic perspective, “dividing and conquering” (DC) and “caching” intermediate results that need to be computed repetitively are two fundamental principles in development of many important algorithms (e.g., dynamic programming [17]). As a matter of fact, the major focus of modern statistical machine learning is to learn (“caching” relevant information) and then carry out inference on top of which [18]. In this review, we provide a brief discussion of important methodological development in molecular modeling as specific applications of these two principles. The content will be organized as follows. Part 2 describes fundamental challenges in molecular modeling; Part 3 summarizes application of these two fundamental algorithmic principles in two lines of methodological research, coarse graining (CG) [18,19,20,21,22,23,24,25,26,27] and enhanced sampling (ES) [28,29,30,31]; Part 4 covers how machine learning, particularly deep learning, facilitates DC and “caching” in CG and ES [29,30,32,33,34,35], Part 5 introduces local free energy landscape (LFEL) approach, a new framework for computational molecular science based on partially transferable in resolution “caching” of local sampling. The first implementation of this new framework in protein structural refinement based on generalized solvation free energy (GSFE) theory [36] is briefly discussed; and Part 6 discusses connections among these three lines of algorithmic development, their specific advantages and prospective explorations. Due to the large body of literature and limited space, we apologize to authors whose excellent works are not cited here.

## 2. Challenges in Molecular Modeling

### 2.1. Accurate Description of Molecular Interactions

Molecular interactions may be accurately described with high level molecular orbital theories (e.g., coupled cluster theory [37,38]) or sophisticated density functionals combined with large basis sets [39,40,41,42,43]. However, such quantum mechanically detailed computation is prohibitively expensive for any realistic complex molecular systems. Molecular interactions are traditionally represented by explicit functions and pairwise approximations as exemplified by typical physics based atomistic molecular mechanical (MM) force fields (FFs) [44,45,46,47]:(1)$$\begin{array}{cc} U(\vec{R}) & =\sum_{bonds}K_{b}{\left(b-b_{0}\right)}^{2}+\sum_{angles}K_{\theta }{\left(\theta -\theta_{0}\right)}^{2}\\ \; & +\sum_{dihedral}K_{\chi }\left(1+cos\left(n\chi -\delta \right)\right)\\ \; & +\sum_{impropers}K_{imp}{\left(\phi -\phi_{0}\right)}^{2}\\ \; & +\sum_{nonbonded}\left(\epsilon_{ij}\left[{\left(\frac{Rmin_{ij}}{r_{ij}}\right)}^{12}-{\left(\frac{Rmin_{ij}}{r_{ij}}\right)}^{6}\right]\right)+\frac{q_{i}q_{j}}{\epsilon r_{ij}} \end{array}$$
or knowledge based potential functions [48,49,50]: these simple functions, while being amenable to rapid computation and are physically sound grounded near local energy minima (e.g., harmonic behavior of bonding, bending near equilibrium bond lengths and bend angles), are potentially problematic for anharmonic interactions, which are very common in some molecular systems [51]. It is well understood that properly parameterized Lennard–Jones potentials are accurate only near the bottom of its potential well. Frustration are ubiquitous in biomolecular systems and are likely fundamental driving force for conformational fluctuations [52,53,54]. One may imagine that a molecular system with all its comprising particles at their respective “happy” energy minima positions would likely be a stable “dead” molecule, which may be a good structural support but is likely not able to provide dynamic functional behavior.

Pairwise approximations are usually adopted for its computational convenience, both in terms of dramatically reduced computational cost and tremendously smaller (when compared with possible many body potentials) number of necessary parameters to be fit in FF parameterization. It is widely acknowledged that construction of traditional FF (e.g., Equation (Equation 1)) is a laborious process. Development of polarizable [9,55,56] and more complex FF with larger parameter set [57] alleviates some shortcomings of earlier counterpart. Expansion based treatments were incorporated to address anharmonicity [58]. However, to tackle limitation of explicit simple functional form and pairwise approximation for better description of molecular interactions remains challenge to be met for molecular modeling community. Additionally, even atomistic simulations are prohibitively expensive for large biomolecular complexes at long time scales (e.g., milliseconds and beyond) [59,60,61].

### 2.2. Inherent Low Efficiency in Sampling of Configurational Space

Complexity of molecular systems is rooted in their molecular interactions, which engender complex and non-linear correlations among molecular degrees of freedom (DOFs). Consequently, effective number of DOFs are greatly reduced. Therefore, complex molecular systems are confined to manifolds [62] of much lower dimensionality with near zero measure in corresponding nominal high dimensional space (NHDS). Consequently, sufficient brute force random sampling in NHDS of interested molecular systems is hopeless.

In stochastic trajectory generation by Monte Carlo (MC) simulations or candidate structural model proposal in protein structure prediction and refinement (or other similar scenarios), new configuration proposal is carried out in NHDS. A lot of effort is inevitably wasted due to sampling outside the actual manifold occupied by the target molecular system. Such wasting may be avoided if we understood all correlations. However, understanding all correlations implicates accurate description of global free energy landscape (FEL) and there is no need to investigate it further. Due to preference of lower energy configurations by typical importance sampling strategies (e.g., Metropolis MC), stochastic trajectories tend to be trapped in local minima of FEL. This is especially true for complex molecular (e.g., biomolecular) systems which have hierarchical rugged FEL with many local minima [63,64]. In trajectory generation by molecular dynamics (MD) simulations, configurational space is explored by laws of classical mechanics and no wasting due to random moves exists. However, molecular systems may well drift away from their true manifolds due to insufficient accuracy of FF, resulting incorrect and wasted sampling. Similar to stochastic trajectory generation, it takes long simulations to map FEL since molecular system tend to staying at any local minimum, achieve equilibrium among many local minima is just as challenging as in the case of stochastic counterpart.

## 3. DC and “Caching” in Traditional Molecular Modeling

To cope with fundamental difficulties in molecular modeling, two distinct lines of methodological development (CG and ES) based on DC and “caching” strategies have been conducted and tremendous progress has been made in understanding of molecular systems. As summarized below:

### 3.1. Coarse Graining, a Partially Transferable “Caching” Strategy

Atomistic FF parameterization is the most well established coarse graining based on time scale separation between each nuclei and its surrounding electrons. Theoretically, MMFFs are potential of mean force (PMF) obtained by averaging over many electronic DOFs for given atomic configurations. In practice, due to the fact that ab initio calculations are expensive and may have significant error when level of theory (and/or basis set) is not sufficient, reference data usually include results from both quantum mechanical (QM) calculations and well-established experimental data [65,66,67]. The DC strategy is utilized by selecting atomic clusters of various size to facilitate generation of QM reference data. The essential information learned from reference data is then permanently and approximately “cached” in FF parameters through the parameterization process. Time scale separation ensures that elimination of electronic DOFs is straight forward but comes with the price of incapability in describing chemical reactions. To harvest benefits of both quantum and atomistic simulations, a well-established DC strategy is to treat a small region involved in interested chemical reaction at QM detail and its surrounding with MMFF [68,69,70,71,72,73]. This series of pioneering work was awarded Nobel prize in 2013, and QM-MM treatment continues to be the mainstream methodology for computational description of chemical reactions [74,75].

The united atom model (UAM) is the next step in coarse graining [76], where hydrogen atoms are merged into bonded heavy atoms. This is quite intuitive since hydrogens have much smaller mass on the one hand, and are difficult to see by experimental detection techniques utilizing electron diffraction (e.g., X-ray crystallography) on the other hand. For both polymeric and biomolecular systems, UAM remains to be expensive for many interested spatial and temporal scales. Therefore, further coarse graining in various forms have been constructed. As a matter of fact, CG is usually used to denote modeling with particles that representing multiple atoms in contrast to atomistic simulations, and the same convention will be adopted in the remaining part of this review unless stated otherwise. Both “Top-down” (that based on reproducing experimental data) and “bottom-up” ( that based on reproducing certain properties of atomistic simulations) approaches are utilized [21,77]. For polymeric materials, beads are either utilized to represent monomers or defined on consideration of persistent length [78], and dissipative particle dynamics (DPD) were proposed to deal with complexities arise from much larger particles [79]. For biomolecular systems, a wide variety of coarse grained models have been developed [20,21,23,24,80,81]. Another important subject of CG methodology development is materials science [82,83]. Earlier definition of CG particles are rather ad hoc [20]. More formulations with improved statistical mechanical rigor appeared later on [22], with radial distribution function based inversion [78,84,85,86], entropy divergence [19] and force matching algorithm [87,88,89] being outstanding examples of systematic development. Present CG is essentially to realize the following mapping as disclosed by Equation (Equation 4) in ref. [22]:(2)$$exp\left[-\beta V_{CG}\left(R_{CG}\right)\right]\equiv \int dr\delta \left(M_{R}\left(r\right)-R_{CG}\right)exp\left[-\beta V(r)\right]$$
with r and R being coordinates in higher resolution and CG coordinates, $M_{R}\left(r\right)$ being the map operator from r to R, *V* and $V_{CG}$ being relevant potential of mean force in higher resolution and CG representation respectively. Due to lack of time scale separation (see Figure 1) for essentially all CG mapping, strict realization of this equation/mapping with exact transferability is not rigorously possible. A naive treatment of CG particles as basic units (with no internal degrees of freedom) would result in wrong thermodynamics [22]. Due to corresponding significant loss of information, it is not possible to develop a definition of CG and corresponding FF parameterization for comprehensive reproduction of atomistic description of corresponding molecular systems. Different coarse graining have distinct advantages and disadvantages, so choosing proper CG strategy is highly dependent upon specific goal in mind. CG particles are usually isotropic larger and softer particles with pairwise interactions, or simple convex anisotropic object (e.g., soft spheroids) that may be treated analytically [24,90,91,92]. Such simplifications provide both convenience of computation and certain deficiency for capturing physics of target molecular systems. CG may be carried out iteratively to address increasingly larger spatial scales by “caching” lower resolution CG distributions with ultra CG (UCG) FF [22,93,94,95,96,97]. Pairwise approximation remains to be limitation of interaction description for traditional CG FF, which may use either explicit simple function form or tables. When compared with atomistic FF, pairwise approximation deteriorate further due to lack of time scale separation (Figure 1).

Another simple and powerful type of CG model for biomolecular systems is Gō model [98,99] and elastic network models (ENM) [100,101,102] or gaussian network models (GNM) [103,104] with native structure being defined as the equilibrium state, and with quadratic/harmonic interactions between all residues within given cutoff. Only a few parameters (e.g., cutoff distance, spring constant) are needed. Such models “caching” the experimental structures and are proved to be useful in understanding major conformational transitions and slow dynamics of many biomolecular systems [105,106].

### 3.2. Enhanced Sampling, a Nontransferable in Resolution DC and “Caching” Strategy

Umbrella sampling (US) [107] is probably the first combination of DC and “caching” strategy for better sampling of molecular system along a given reaction coordinate (RC) (or order parameter) *s*. DC strategy is first applied by dividing *s* into windows, information for each window is then partially “cached” by corresponding bias potentials and local statistics. Later on, adaptive US (AUS) [108,109] and weighted histogram analysis method (WHAM) [110] were developed to improve both efficiency and accuracy. MBAR [111,112] was developed to achieve error bound analysis which was not available in WHAM. Further development including adaptive bias force (ABF) [113,114,115] and metadynamics [116,117,118]. Details of these methodologies were well explained by excellent reviews [119,120,121,122]. The common trick to all of these algorithms (and their variants) is to “cache” visited configurational space with bias potentials/force and local statistics, thus reduce time spent in local minima and dramatically accelerate sampling of interested rare events. Denote CV as s(r) (r being physical coordinates of atoms/particles in the target molecular system), equilibrium distribution and free energy on the CV may be expressed as [30]:(3)$$\begin{array}{cc} p_{0}\left(s\right) & =\int dr\delta \left[s-s(r)\right]p_{0}\left(r\right)=\langle \delta \left[s-s(r)\right]\rangle \end{array}$$(4)$$\begin{array}{cc} p_{0}\left(r\right) & =\frac{e^{-\beta U(r)}}{\int dre^{-\beta U(r)}} \end{array}$$(5)$$\begin{array}{cc} F(s) & =-\frac{1}{\beta }log\left[p_{0}\left(s\right)\right] \end{array}$$(6)$$\begin{array}{cc} F(s) & =-\frac{1}{\beta }log\left[p\left(s\right)\right]-V\left(s\right) \end{array}$$
with p(s) being the sampled distribution in simulations with corresponding bias potential V(s) for “caching” of visited configurational space.

The starting point of these “caching” algorithms is specification of reaction coordinates (RC) or collective variables (CVs), which is a very challenging task for complex molecular systems in many cases. Traditionally, principle component analysis (PCA) [123] is the most widely utilized and a robust way for disclosing DOFs associated with the largest variations. To deal with ubiquitous nonlinear correlations, kernels are often used albeit with the difficulty of choosing proper kernels [124]. Additional methodologies, include multidimensional scaling (MDS) [125], isomap [126], locally linear embedding (LLE) [127], diffusion map [128,129] and sketch map [130] have been developed to map out manifold for high dimensional data. However, each has it own limitations. For example, LLE [127] is sensitive to noise and therefore has difficulty with molecular simulation trajectories which are quite noisy; Isomap [126] requires relatively homogeneously sampled manifold to be accurate. Both LLE and Isomap do not provide explicit mapping between molecular coordinates and CVs; diffusion and sketch maps are likely to be more suitable to analyze molecular simulation trajectories. Nonetheless, their successful application for large and complex molecular systems remains to be tested. All of above non-linear mapping algorithm are mainly suitable for manifold on a single scale, and capturing manifold on multiple scales simultaneously in molecular simulations has not been reported yet. When we are interested in finding paths for transitions among known metastable states, transition path sampling (TPS) [131,132,133] methodology maybe utilized to establish CV.

Apparently, RC and/or CV based ES is a different path for facilitate simulation of complex molecular systems on longer time scales from coarse graining. One apparent plus side is that these algorithms are “in resolution” as no systematic discarding of molecular DOFs occur. With specification of RC and/or CVs, computational resource is presumably directed toward the most interesting dynamics of the target molecular system, and RC and/or CV maybe repetitively refined to obtain mechanistic understanding of interested molecular processes. However, the down side is that “cached” information on local configurational space is not transferable to other similar molecular systems. While rigorous transferability may not be easily established for any CG FF, practical utility of CG FF for molecular systems with similar composition and thermodynamic conditions have been quite common and useful [24]. Therefore, CG FF may be deemed as partially transferable.

An important recent development of DC strategy for enhanced sampling is Markov state models (MSM) [134,135,136,137], one great advantage of which is that no RC or CV is needed. Instead, it extracts long-time dynamics from independent short trajectories distributed in configurational space. Many important biomolecular functional processes have been characterized with this great technique [138,139,140]. The most fundamental assumption is that all states for a target molecular system form an ergodic Markov chain:(7)π(t+τ)=π(t)P
with π(t) and π(t+τ) being a vector of probabilities for all states at time *t* and t+τ respectively. P is the transition matrix with its element $P_{ij}$ being probability of the molecular system being found in state *j* after an implied lag time (τ) from the previous state *i*. Apparently as *t* goes to infinity for an equilibrium molecular system, a stationary distribution π will arise as defined below:(8)π=πP

The advantage of not needing RC/CV does not come for free but with accompanying difficulties. Firstly, one has to distribute start points of trajectories to statistically important and different part of configurational space, then select proper (usually hierarchical, with each level of hierarchy corresponds to a specific lag time) partition of configurational space into discrete states. This is the key step of DC strategy in MSM. No formal rule is available and experience is important. In many cases some try and error is necessary. Secondly, within each discrete state at a given level of hierarchy, equilibration is assumed to be achieved instantly and this assumption causes systematic discretization error, which fortunately may be controlled with proper partition and sufficiently long lag time [141]. Apparently, metastable states obtained from MSM analysis is molecular system specific and thus not transferable.

Another important class of enhanced sampling is to facilitate sampling with non-Boltzmann distributions and restore property at targeted thermodynamic condition through proper reweight [142]. Most outstanding examples are Tsallis statistics [143,144], parallel tempering [145,146], replica exchange molecular dynamics [147,148], Landau-Wang algorithm [149] and integrated tempering sampling (ITS) [150,151,152]. These algorithms are not direct applications of DC and “caching” strategies and are not discussed further here.

## 4. Machine Learning Improves “Caching”

### 4.1. Toward Ab Initio Accuracy of Molecular Simulation Potentials

Fixed functional form and pairwise approximation of non-bonded interactions are two major factors limiting the accuracy of molecular interaction description in both atomistic and some CG FF. Neural network (NN) has capability of approximating arbitrary functions and therefore has potential to address these two issues. Not surprisingly, significant progress has been made in this regard as summarized by recent excellent reviews [27,153,154,155,156,157]. Cutoff and attention to local interactions remains the DC strategy for development of machine learning potentials. The major improvement over traditional FF is better “caching” that overcomes pairwise approximation and fixed functional form limitations. NN FF naturally tackle both issues as explicit functions are not necessary since NNs are universal approximators. The significance of many-body potentials [158] and extent of pairwise contributions were analyzed [159,160]. It is important to note that despite the fact that pairwise interactions account for the majority of energy contributions, high ordered interactions are likely to be significant in shaping differences of subtly distinct molecular systems. There are also efforts to search for different and proper simple functional forms, which are expected to be more accurate than present functional forms in traditional FF on the one hand, and alleviate overfitting/generalization difficulty and reduce computational cost of complex NN FF on the other hand [161,162], especially when training dataset is small. While most machine learning FF are trained by energy data [153,157], gradient-domain machine learning (GDML) approach [163] directly learns from forces and realizes great savings of data generation.

Just as in the case of traditional FF, transferability and accuracy is always a tradeoff. More transferability implicates less attention is paid to “cache” detailed differences among different molecular systems, hence less accuracy. Exploration in this regard, however, remains not as much as necessary [164,165,166]. Unlike manual fitting of traditional FF, systematic investigation of tradeoff strategies is potentially feasible for machine learning fitting [167], and yet to be done for many interesting molecular systems. With expediency of NN training, development of a NN FF hierarchy with increasing transferability/accuracy and decreasing accuracy/transferability is likely to become a pleasing reality in the near future. Rapid further development of machine learning potentials, particularly NN potentials, are expected. However, significant challenges for NN potentials remain on better generalization capability, description/treatment of long range interactions [168,169], wide range of transferability, [170] faster computation [171] and proper characterization of their error bounds. Should further significant progress be made on these issues, it is promising we may have routine molecular simulations with both classical efficiency and ab initio accuracy in the near future.

### 4.2. Machine Learning and Coarse Graining

As in the case of constructing atomic level potentials, machine learning has been applied to address two outstanding pending issues in coarse graining, which are definition of CG sites/particles and parameterization of corresponding interactions between/among these sites/particles. Traditional CG FF, suffers from both pairwise approximation and, for some, accuracy ceiling of simple fixed functional forms which are easy to fit. By using more complex (but fixed functional form) potentials with a machine learning fitting process, Chan et al. [172] developed ML-BOP CG water model with great success. Deep neural network (DNN) was utilized to facilitate parameterization of CG potentials when given radial distribution functions (RDF) from atomistic simulations [173]. CGnet demonstrated great success with simple model systems (alanine dipeptide) [174]. DeePCG model was developed to overcome pair approximation and fixed functional form and demonstrated with water [175]. Using oxygen site to represent water is rather intuitive. However, for more complex biomolecules such as proteins, possibility for selection of CG site explodes. To improve over intuitive or manual try and error definition of CG sites, a number of studies have been carried out [176,177,178,179] to provide better and faster options for choosing CG sites. However, no consensus strategy is available up to date and more investigations are desired. The fundamental difficulty is that there is no sufficient time scale separation between explicit CG DOFs and discarded implicit DOFs, regardless of specific selection scheme being utilized. Intuitively, one would expect CG FF parameters to be dependent upon definition of CG sites/particles. In this regard, auto-encoders were utilized to construct a generative framework that accomplishes CG representation and parameterization in a unified way [33]. The spirit of generative adversarial networks was utilized to facilitate CG construction and parameterization, particularly with virtual site representation [180]. It was found that description of off-target property by CG exhibit strong correlation with CG resolution, to which on-target property being much less sensitive [181]. Such observation suggests that adjust CG for specific target properties might be a better strategy than searching for a single best CG representation. Despite potentially more severe impact of pairwise approximation for CG FF than in atomistic FF, quantitative analysis in this regard remain to be done to the best of our knowledge.

### 4.3. Machine Learning in Searching for RC/CVs and Construction of MSM

To overcome difficulties of earlier nonlinear CV construction algorithms [126,127,128,130] and to reduce reliance on human experience, auto-encoders, which is well-established for trainable (non-linear) dimensionality reduction, are utilized in a few studies [182,183,184,185]. Chen and Ferguson [183] first utilized autoencoders to learn nonlinear CVs that are explicit and differentiable functions of molecular coordinates, thus enabling direct utility in molecular simulations for more effective exploration of configurational space. Further improvement [182] was achieved through circular network nodes and hierarchical network architectures to rank-order CVs. Wehmeyer and Noé [184] developed time-lagged auto-encoder to search for low dimensional embeddings that capture slow dynamics. Ribeiro et al. [185] proposed the reweighted autoencoded variational Bayes to iteratively refine RC and demonstrated in computation of the binding free energy profile for a hydrophobic ligand-substrate system. Building a MSM for any specific molecular system requires tremendous experience and many steps in process are error prone. To overcome these pitfalls, VAMPnet that based on variational approach for Markov process was developed to realize the complete mapping steps from molecular trajectories to Markov states [186]. As physical understanding of interested molecular systems is essential and the ultimate goal, application of these methods as black boxes are not encouraged.

## 5. The Local Free Energy Landscape Approach

Both CG and ES methodologies facilitate molecular simulation by effectively reducing local sampling. In CG, it is realized through “caching” (integration) of distributions for faster/discarded DOFs with proper CG FF, and thus has the inevitable cost of losing resolution (information), accompanied by the desired attribute of (partial) transferability to various extent. ES reduces lingering time of molecular systems in local minima through “caching” visited local configurational space, which is usually defined by relevant DC strategies, with biasing potentials. When compared with CG, there is no resolution loss. However, “cached” manifold of configurational space is molecular process specific and thus not transferable at all. In molecular modeling community, these two lines of methodologies are developed quite independently. Nevertheless, one might want to ask why not have both advantages in one method, that is to reduce repetitive local sampling without loss of resolution and with “cached” results being partially transferable. The local free energy landscape (LFEL) approach [187] is proposed with this intention in mind. Historically, parameterization of FF by coarse graining has been the only viable framework due to two fundamental constraints. Firstly, in earlier days of molecular modeling, typical computers have memory space of megabytes or less, render it impossible to accommodate millions or more parameters needed to fit complex LFEL; secondly, while both neural network and autodifferentiation were invented decades ago, the computational molecular science community did not master these techniques for fitting large number of parameters efficiently until recently. With these two constraints removed, possibility for alternative path arise to break monopoly of classical molecular modeling by FF parameterization via coarse graining. Specifically, one may carry out direct fitting of LFEL and all important information on local distributions of molecular DOFs obtained from expensive local sampling may be “cached”. This is in strong contrast to coarse graining based parameterization, in which local distributions are substituted by averaging in relevant lower dimensional space projection (e.g., pairwise distances among CG sites). However, it is essential to assemble LFEL and construct FEL of the interested molecular system, and this is the core of the LFEL approach. For a molecular system with *N* DOFs, this LFEL approach may be expressed as:(9)$$P\left(r_{1},r_{2},\cdots ,r_{N}\right)=P\left(R_{1},R_{2},\cdots ,R_{M}\right)\left(M\leq N\right)$$
(10)$$R_{i}=\left(r_{i_{1}},r_{i_{2}},\cdots ,r_{i_{l}}\right)$$
(11)$$\begin{array}{c} P(R_{1},R_{2},\cdots ,R_{M})\\ =\prod_{i=1}^{M}P\left(R_{i}\right)\frac{P(R_{1},R_{2},\cdots ,R_{M})}{\prod_{i=1}^{M}P\left(R_{i}\right)} \end{array}$$
(12)$$\begin{array}{cc} \; & \approx \prod_{i=1}^{M}P\left(R_{i}\right),\mathrm{and}\;\mathrm{sampling}\;\mathrm{all}\;Rs\;\mathrm{with}\;\mathrm{mediated}\;\mathrm{GCF} \end{array}$$
(13)$$\begin{array}{cc} G & =-k_{B}TlnP\left(r_{1},r_{2},\cdots ,r_{N}\right)\\ \; & =-k_{B}TlnP\left(R_{1},R_{2},\cdots ,R_{M}\right)\\ \; & \approx -k_{B}T\sum_{i}^{M}lnP\left(R_{i}\right),\mathrm{and}\;\mathrm{sampling}\;\mathrm{all}\;Rs\;\mathrm{with}\;\mathrm{mediated}\;\mathrm{GCF} \end{array}$$
an *N*-DOF molecular system is reorganized into *M* overlapping regions (Equation (Equation 9)), each region has some number of DOFs (Equation (Equation 10)). The key step of LFEL approach is expressed in Equation (Equation 11), in which the first product term (addressed as “local term(s)” hereafter) treat *M* regions as if they were independent, and all correlations among different regions are incorporated by the fraction term, which is termed global correlation fraction (GCF) and is extremely difficult, if ever possible, to be calculated directly. However, GCF is a unnormalized probability distribution, when all molecular DOFs in local terms are (approximately) sampled according to GCF, then we do not need GCF explicitly anymore (Equations (12) and (Equation 13)). GCF represents two types of global correlations. The first type is mediated correlations among different regions by the fact that they overlap, and relevant molecular DOFs in such overlapping space shared by different regions should have exact same state for all concerning regions. The second type is direct global correlations among molecular DOFs in different regions caused by genuine long-range interactions (e.g., electrostatic interactions). Satisfying the first type with sampling is trivial, and ensuring all overlapping regions share the exact same state is sufficient (Equations (12) and (Equation 13)). The second type of global correlations need more involved treatment. These equations are apparently of general utility for any multiple-variable (high-dimensional) problem. In the specific case of a complex molecular system, using one set of coordinates realizes the mediated contribution of GCF. The approximation in Equation (12) is made by ignoring the second type of global correlations. Free energy minimization of a molecular system in thermodynamic equilibrium may be treated as maximization of joint probability (Equation (Equation 13)). For molecular systems (or biological systems) off equilibrium, the joint distribution remains our focus despite free energy is not well defined anymore. A schematic representation of the LFEL approach in contrast to FF framework is shown in (Figure 2). While we only demonstrated GSFE implementation of LFEL at residue level for protein structural refinement. LFEL approach may be utilized to “cache” local distributions at any spatial scales. Just as there are many methodological developments in the mainstream FF framework, there are certainly many possible ways to develop algorithms in the LFEL approach. We explored a first step toward this direction through a neural network implementation of the generalized solvation free energy (GSFE) theory [36]. In GSFE theory, each comprising unit in a complex molecular system is solvated by its neighboring units. Therefore, each unit is both a solute itself and a comprising solvent unit of its solvent units. Let $\left(x_{i},y_{i}\right)=R_{i}$ denote a region *i* defined by a solute $x_{i}$ and its solvent $y_{i}$, a molecular system of *N* units has *N* overlapping regions. Each local term may be further expanded:(14)$$\begin{array}{cc} P(R_{i}) & =P(x_{i},y_{i})\\ \; & =P\left(x_{i}|y_{i}\right)P\left(y_{i}\right) \end{array}$$

Both terms may be learned from either experimental or computational datasets, as long as they are sufficiently representative and reliable. The first term in Equation (Equation 14) is the likelihood term when $x_{i}$ is the given, it quantifies the extent of match between the solute $x_{i}$ and its solvent $y_{i}$. The second term is the local prior term, it quantifies the stability of the solvent environment $y_{i}$. Computation of the prior term is more difficult than the likelihood term, but certainly learnable when sufficient data is available. A local maximum likelihood approximation of GSFE (LMLA-GSFE) is to simply ignore local prior terms.

A particular implementation of the LMLA-GSFE for protein structure refinement with residues defined as comprising unit was conducted [187]. In this scheme, GSFE is integrated with autodifferentiation and coordinate transformation to construct a computational graph for free energy optimization. With fully trainable LFEL derived from backbone and $C_{\beta }$ atom coordinates of selected experimental protein structures, we achieved superb efficiency and competitive accuracy when compared with state of the art atomistic protein refinement refinement methodologies. With our newly developed pipeline, refinement of typical protein structure decoys (within 300 amino acids) takes a few seconds on a single CPU core, in contrast to a few hours by typical efficient sampling/minimization based algorithms (e.g., FastRelax [188]) and thousands of hours for MD based refinement [189]. In the latest CASP14 refinement contest (predictioncenter.org/casp14/index.cgi (accessed on 15 February 2021)), our method ranked the the first for the 13 targets with start GDT-TS score larger than 60. We expect incorporation of complete heavy atom information and local prior terms to further improve this method in the future. GSFE theory in particular and the LFEL approach in general, are certainly extendable to modeling of other soft matter molecular systems.

## 6. More on Connections among CG, ES and LFEL Approach

All of these algorithms have a common goal of accelerating computation of a joint distribution for a given molecular system at some target resolution, albeit from distinct perspectives. The fundamental underpinning is the fact that molecular correlations among its various DOFs limit a molecular system to a manifold of significantly lower dimension. Both ES and CG in the FF framework and the LFEL approach are distinct strategies to “cache” manifolds from either configurational space (Figure 3) or physical space perspective (Figure 4). Commonality and differences of these strategies are summarized in Table 2 and discussed below.

Both MSM and RC/CV based ES are designed to first describe local parts of the approximate manifold in the configurational space formed by all molecular DOFs of the target molecular system. Information for such local configurational space is partially “cached” either as bias potentials or transition counts, which are further processed to map FEL and dynamics of interested molecular processes. Computational process (or educated guess) for establishment of RC/CVs is essentially “caching” results from sampling/guessing local parts of the configurational as approximate relevant manifold (Figure 3B). Subsequent sampling along RC/CV is hoped to disclose our interested molecular processes (e.g., biomolecular conformational transitions, substrate binding/release in catalysis). Involved molecular DOFs for RC/CVs are not necessarily spatially adjacent on the one hand, and may be different for different molecular processes of the same molecular system. Apparently, RC/CVs are molecular process specific and not transferable, even among different molecular process of the same molecular system. Nonetheless, the methodology for searching CVs may be applied to many different molecular processes/systems.

In contrast, both CG in the FF framework and the LFEL approach are motivated to “cache” relevant information on the complete configurational distribution for local clusters of molecular DOFs. Such local clusters are building blocks for many similar molecular systems (e.g., AAs in protein molecular systems) and consequently have limited and approximate transferability. In CG, strongly correlated local clusters of molecular DOFs are represented as a single particle, complex many body correlations/interactions of CG particles within selected cutoff distances are represented by simplified CG FF in a lower resolution and longer range correlations/interactions are incorporated either through more coarser CG models or by separate long-range interaction computation. In LMLA-GSFE implementation of LFEL, all complex many body correlations within selected regions (i.e., each solute and its specific solvent) are decomposed into two terms in Equation (Equation 14), local likelihoods and local priors in the same resolution, with local priors and direct genuine long-range interactions simply ignored, and LFEL being approximated by local likelihood terms. More and better ways for implementing LFEL are expected in the future.

The first step of CG is to partition atoms/particles of high resolution representation into highly correlated local clusters that will be represented by corresponding single CG particles, and moderately correlated regions define interaction cutoff for CG particles; The second step is to select a site (usually one of the comprising high resolution particles) to represent the corresponding highly correlated cluster; The third step is to select functional forms to describe molecular interactions among newly defined CG particles, and parameters are optimized by selected loss functions (e.g., differences of average force in force matching [87,88]) based on sampling in the whole configurational space of molecular systems and hopefully to be transferable to some extent. One may imagine that both best clustering and optimal representation sites of clusters may vary with different functional forms used to describe CG particle interactions and in different part of configurational space. Neural network based CG potentials do not have limitation of fixed functional form and pairwise approximations. However, the need to partition molecular systems into transferable clusters and to specify representation site/particle remain. For all different forms of CG, the fundamental essence is to “cache” many body potential of mean force (PMF) in simplified CG FF at a lower resolution. In contrast, LFEL approach is to first using a DC strategy to divide molecular systems into local regions, then directly “cache” many body PMF (or LFEL) of such local regions in the original resolution. The cached complex local multivariate distributions in NN are subsequently utilized to construct FEL of target molecular system through dynamic puzzle assembly based on sampling with GCF as expressed in Equations (12) and (Equation 13). In language of statistical machine learning. Training of LFEL is the learning step, while construction of global FEL is the inference step. The advantage of CG is a simpler resulting physical model, but is inflexible due to fixed clustering and representation on the one hand, and lost resolution/information on the other hand. Properly implemented LFEL while has selected spatial regions comprising many molecular DOFs, composition of such regions are fully dynamic. For example, in GSFE implementation of LFEL, a region is defined by a solute unit and all of its solvent units, and comprising units for the solvent is dynamically updated in each iteration of free energy optimization. Additionally, no loss of resolution is involved for LFEL approach. Hence all difficulties and uncertainties associated with molecular DOF partition, CG site selection and time scale separation, all of which apparently limit transferability of CG FF, disappear. Correspondingly, the extent of transferability of a LFEL model is in principle at least no worse than CG FF. Differences of CG and specific implementation of LFEL by GSFE theory is schematically illustrated in Figure 4. The superior efficiency of LFEL approach comes with a price. The assembled global FEL has arbitrary unit for two reasons. Firstly, it is extremely difficult to obtain the partition function (normalization constant) for local regions directly during the training/caching stage, therefore we effectively obtain the LFEL up to an unknown constant. Secondly, for two different molecular systems, the number of local regions are usually different and so is the corresponding normalization constant.

These three lines of algorithms may be combined to facilitate molecular modeling. For example, one might first utilize deep learning based near quantum accuracy many body FF to perform atomistic simulations for protein molecular systems, and then extracting local distributions properly with some form of LFEL, which may potentially be utilized to simulate protein molecular systems with near-quantum accuracy and at regular amino-acid based CG or even much faster speed! Similarly, one may extract and “cache” large body of information from residue level CG simulations with proper LFEL implementation, which may be utilized to achieve ultra CG (UCG) efficiency with residue resolution. Application of CV and MSM based ES algorithm for CG models is straight forward. Combination of LFEL with CV or MSM based ES is more subtle and yet to be investigated.

## 7. Conclusions and Prospect

The application of “dividing and conquering” and “caching” principle in development of molecular modeling algorithms is briefed. Historically, coarse graining and enhanced sampling have been two independent lines of methodological development in the mainstream FF framework. While they share the common goal of reducing local sampling, the formulations are completely different with distinct (dis)advantages. Coarse graining obtains partial transferable FFs but loses resolution, enhanced sampling retains resolution but results are not transferable. The LFEL approach suggests a third strategy to directly approximate global joint distribution by superposition of LFEL, which may be learned from available dataset of either experimental or computational origin. Through integration of coordinate transformation, autodifferentiation and neural network implementation of GSFE, our recent work of protein structure refinement demonstrated that simultaneous realization of transferable in-resolution “caching” of local sampling is not only feasible, but also highly efficient due to replacement of local sampling by differentiation. It is hoped that this review stimulates further development of better “dividing and conquering” strategies for complex molecular systems through more elegant, efficient and accurate ways of “caching” potentially repetitive computations in molecular modeling at various spatial and temporal scales. With diverse molecular systems (e.g., nanomaterials, biomolecular systems), specialization of methodology is essential to take advantage of distinct constraints and characteristics.

## Figures and Tables

**Figure 1 ijms-22-05053-f001:**
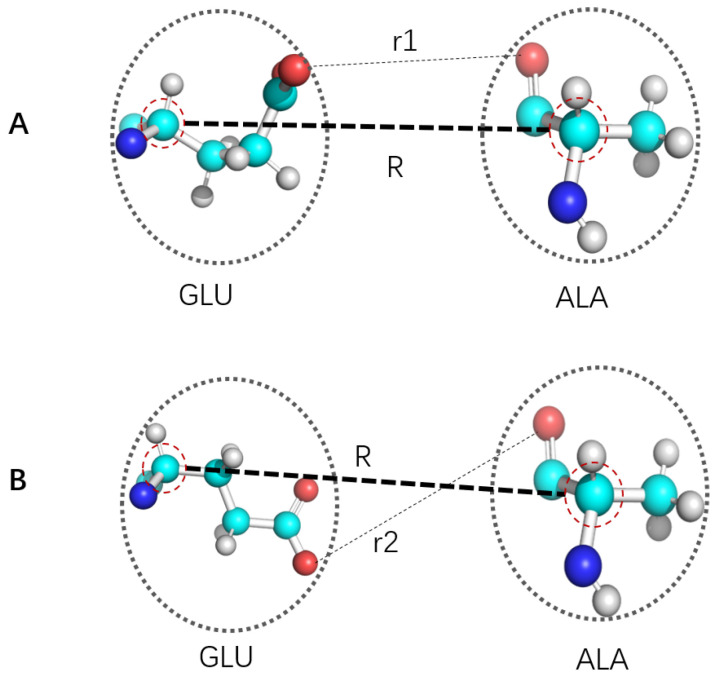
Schematic illustration of time scale separation issue in CG. (**A**,**B**) show two situations with $C_{\alpha }$ distances between two amino acids GLU and ALA being *R*, but with GLU have different conformations. If $C_{\alpha }$ atoms were defined as CG site, then these two relative conformation with distinct interactions would be treated as the same. In (**A**,**B**), CG site distance in both (**A**,**B**) are *R*, but many other pairs of atoms have distinct distances as exemplified by $r_{1}$ and $r_{2}$. Such treatment would only be true if for any small amount of displacement of $C_{\alpha }$, side chains accomplished many rotations and thus may be accurately represented by averaging (i.e., with good time scale separation). This issue is apparently not limited to the specific definition of $C_{\alpha }$ being CG site, but rather general for essentially all CG development.

**Figure 2 ijms-22-05053-f002:**
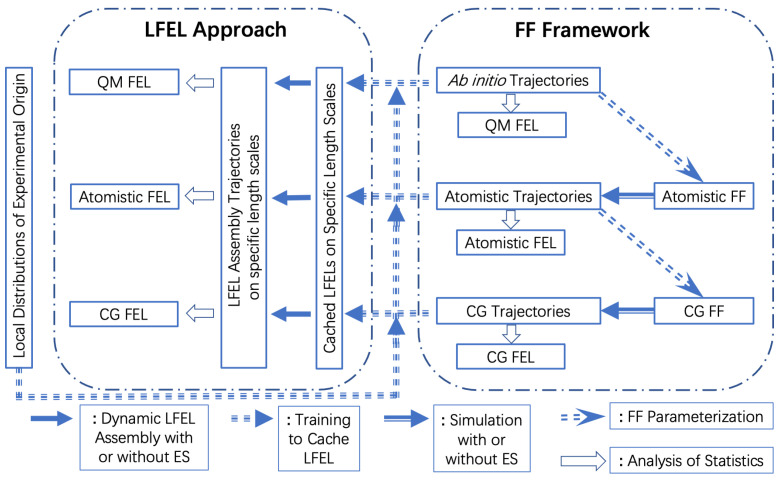
Schematic illustration of the LFEL approach in contrast to present mainstream FF framework. FF parameterization is the foundation for present classical computational molecular science. Training of neural network for “caching” LFEL is the foundation for LFEL approach, the source data can be either of experimental or computational origin. In FF framework, simulation (with or without ES) is driven by FF, in LFEL approach, propagation of molecular systems to minimize free energy (or maximize joint probability) is driven by compromise among many LFELs. Expensive repetitive local sampling in FF framework is substituted by differentiation w.r.t. LFELs.

**Figure 3 ijms-22-05053-f003:**
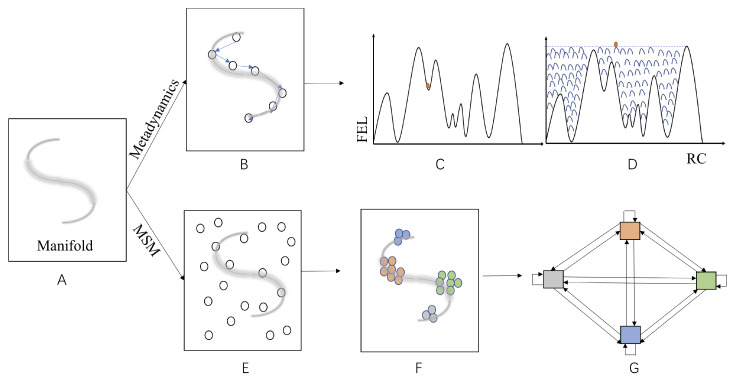
Schematic illustration of essential features for enhanced sampling by metadynamics and MSM. (**A**) The “S” shape grey line represents the unknown manifold in the configurational space (represented by the square) of a molecular system. (**B**) Small circles connected by blue arrows represent computed (guessed) RC/CVs for the molecular system, which is utilized to conduct metadynamics simulations. (**C**) The FEL of the molecular system along the computed/selected RC/CV in (**B**,**D**) “Caching” of the LFEL by bias potentials (gaussians represented by blue bell shaped lines) accumulated in the course of metadynamics simulations. (**E**) Distribution of the molecular system to the whole configurational space at the start of a MSM simulation, small circles represent initial start points for short MSM trajectories. (**F**) Sampling results of short MSM trajectories fall mainly near the manifold, distinct “states” are represented by different colors. (**G**) Establishment of transition matrix by transition counts between “states” obtained from short trajectories.

**Figure 4 ijms-22-05053-f004:**
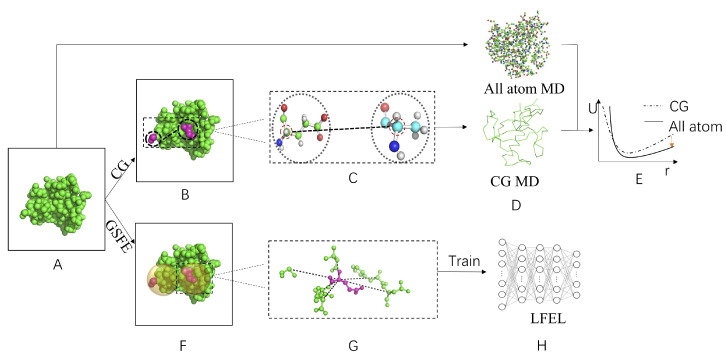
Schematic illustration of difference between CG and GSFE implementation of LFEL using protein as an example. (**A**) Target molecular systems in physical space. Due to the goal of constructing partially transferable models and/or force fields, usually many different but similar molecular systems are considered. (**B**) Selection of local atom/particle clusters to be represented as one particle in CG model. (**C**) Selection of CG sites. (**D**) Comparison between atomistic (or higher resolution) simulation results and CG (lower resolution) results. (**E**) Adjust of CG FF parameter according to comparison from (**D**). (**F**) Definition of solvent region for each solute unit. (**G**) Feature extraction for each solute. (**H**) “Caching” of LFEL with neural network by training with prepared data sets.

**Table 1 ijms-22-05053-t001:** Number of publications from Web of Science search on 8 September 2020.

Key Words	Number of Publications
Molecular dynamics simulation	241,748
Monte Carlo simulation	189,550
QM-MM (quantum mechanical—molecular mechanical) simulation	9907
Dissipative particle dynamics simulation	3693
Langevin dynamics simulation	3893
Molecular modeling	2,072,091
All of the above	2,243,182

**Table 2 ijms-22-05053-t002:** Commonality and difference among three types of algorithms.

Algorithm	Coarse Graining	Enhanced Sampling	LFEL Approach
Resolution	Lower	In	In
Transferable?	Partial	No	Partial
Dividing space	Physical	Configurational	Physical
Free energy unit	Partially Specified	Specified	Arbitrary

## Data Availability

Data sharing not applicable.

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
