# Peer review of "“Dividing and Conquering” and “Caching” in Molecular Modeling"

_ijms, 2021, doi:10.3390/ijms22095053_

Round 1
Reviewer 1 Report
In the letter to the Editor, the authors wrote that the review is sent to RSC Advances and not to the International Journal of Molecular Sciences. This must be clarified as the article should not be sent to two different journals at the same time.
The review is interesting and discusses important challenges in molecular modeling: an accurate description of molecular interaction and efficient sampling methods. It presents different approaches to coarse-graining molecular systems. The recent progress in the machine learning approach to molecular potentials and sampling is also discussed. Finally, the local free energy landscape approach is presented as another strategy
that facilitates molecular modeling through partially transferable in resolution caching of distributions for local clusters of molecular degrees of freedom.
The review cites almost 200 references, mostly from the last few years which is adequate.
Author Response
We carefully revised the whole manuscript and detailed changed are highlighted. Please see the attachment.

Reviewer 2 Report
In this review, Cao and Tian focused on reviewing the main methodological developments in molecular modeling in the past years, including the most recent innovations in the adaptation of machine learning related methods to this area of research. Overall, this review gives a broad view of the several methods, without focusing too much on the methodological details, but with an introductory perspective that can easily be followed by new researchers. The approach used by the authors to move away from the detailed description of the more traditional methods, focusing more on the new trendy machine learning approaches is something not so much seen in other reviews. The review is well written with some pin-pointed grammar errors that should be corrected. I found this review very interesting and, in my opinion, it can follow on for publication after a thorough revision of the english.
Author Response
We made changes throughout the manuscript for both English issues and more accurate representation of ideas. Modifications are highlighted in the revised manuscript. Please see the attachment.
